# Markers Useful in Monitoring Radiation-Induced Lung Injury in Lung Cancer Patients: A Review

**DOI:** 10.3390/jpm10030072

**Published:** 2020-07-26

**Authors:** Mariola Śliwińska-Mossoń, Katarzyna Wadowska, Łukasz Trembecki, Iwona Bil-Lula

**Affiliations:** 1Department of Medical Laboratory Diagnostics, Division of Clinical Chemistry and Laboratory Haematology, Wroclaw Medical University, ul. Borowska 211A, 50-556 Wroclaw, Poland; mariola.sliwinska-mosson@umed.wroc.pl (M.Ś.-M.); iwona.bil-lula@umed.wroc.pl (I.B.-L.); 2Department of Radiation Oncology, Lower Silesian Oncology Center, pl. Hirszfelda 12, 53-413 Wroclaw, Poland; lukasz.trembecki@umed.wroc.pl; 3Department of Oncology, Faculty of Medicine, Wroclaw Medical University, pl. Hirszfelda 12, 53-413 Wroclaw, Poland

**Keywords:** lung cancer, radiotherapy, radiotherapy monitoring, radiation-induced lung injury, RILI, pneumonitis, radiation-induced lung fibrosis, RILF, circulating biomarkers, microRNA

## Abstract

In 2018, lung cancer was the most common cancer and the most common cause of cancer death, accounting for a 1.76 million deaths. Radiotherapy (RT) is a widely used and effective non-surgical cancer treatment that induces remission in, and even cures, patients with lung cancer. However, RT faces some restrictions linked to the radioresistance and treatment toxicity, manifesting in radiation-induced lung injury (RILI). About 30–40% of lung cancer patients will develop RILI, which next to the local recurrence and distant metastasis is a substantial challenge to the successful management of lung cancer treatment. These data indicate an urgent need of looking for novel, precise biomarkers of individual response and risk of side effects in the course of RT. The aim of this review was to summarize both preclinical and clinical approaches in RILI monitoring that could be brought into clinical practice. Next to transforming growth factor-β1 (TGFβ1) that was reported as one of the most important growth factors expressed in the tissues after ionizing radiation (IR), there is a group of novel, potential biomarkers—microRNAs—that may be used as predictive biomarkers in therapy response and disease prognosis.

## 1. Introduction

According to the WHO (2018), cancer is a leading cause of death worldwide, accounting for an estimated 9.6 million deaths. Lung cancer, which was rare before 1900 with fewer than 400 cases described in the medical literature, now is the most common cancer, accounting for an 2.09 million cases in 2018, and the most common cause of cancer death (1.76 million deaths) [1,2,3]. Lung cancer is regarded as any tumor of the respiratory epithelium or pneumocytes, whose main risk factors are exposure to environmental carcinogens, irradiation, and genetic disorders [4,5]. Clinical classification distinguishes two main groups of lung cancer, small cell lung carcinoma (SCLC, 15% of all lung cancers) and non-small cell lung carcinoma (NSCLC, 85% of all lung cancers), which are additionally subcategorized into adenocarcinoma, squamous cell carcinoma (SCC), and large cell carcinoma [6,7,8]. The distinction between adenocarcinoma and SCC is crucial for therapeutic decision making. The most significant dividing line is between patients who are candidates for surgical lung cancer resection and inoperable patients who will benefit from chemotherapy, radiotherapy (RT) or both [3]. The management of therapeutic strategies depends on lung cancer stage at the time of its diagnosis and is listed in Table 1.

RT has a potential role in all stages of NSCLC, and it can be used either as definitive or palliative therapy. RT should be provided for all patients with stage III NSCLC and in patients with stage IV disease who may benefit from local therapy. Uses of RT for NSCLC include (1) definitive therapy for locally advanced NSCLC, generally combined with chemotherapy; (2) definitive therapy in patients with early-stage disease, who are medically inoperable, who refuse surgery, or who are high-risk surgical candidates; (3) preoperative or postoperative therapy for selected patients treated with surgery; (4) therapy for limited recurrences and metastases; and (5) palliative therapy for patients with incurable NSCLC [9,10,11,12,13]. The main restrictions of RT are tumor hypoxia, repopulation, DNA damage repair, and molecular mechanisms linked to RT resistance and treatment toxicity, manifesting in radiation-induced lung injury (RILI). RILI, next to the local recurrence and distant metastasis are substantial challenges to the successful management of lung cancer [2,13,14,15,16].

Contrary to the increase in number of lung cancer cases over the last few decades, the 5-year overall survival (OS) of lung cancer patients has not changed appreciably. Despite recent advances in understanding the molecular biology of lung cancer and the introduction of new therapeutic agents in the treatment, the 5-year OS rate is less than 16% [12,13,17]. These data indicate an urgent need for the development of novel, more personalized therapeutic approaches and more precise markers (molecular biomarkers for radiosensitivity, immune host markers, fuller imaging analysis like radiomics) of individual response and risk of side effects. Improvement of the RT’s efficacy requires maximization of tumor control and minimization of RT treatment toxicity [9,11,14].

Regardless of a vast number of cytokines, growth factors and circulating markers representing potential culprit biological processes that are involved in radiotherapy response, there are no blood-based biomarkers in clinical practice that would enable assessment of their relationship with radiotherapy response and toxicity. Current data in preclinical practice report a possible application of microRNA as radiosensitizing biomarkers in RT response and prognosis [13,19]. In this study, we provide an overview of possible markers useful in monitoring RILI in lung cancer patients, starting with circulating cytokines pro- and anti-inflammatory and ending with the molecular approaches in the diagnostics.

## 2. Radiation-Induced Lung Injury

Generally, RT is a low-toxic treatment; however, the lung is a radiosensitive organ and tends to be easily damaged by radiation beams. Depending on the assessment methods, it is estimated that about 5% to nearly 40% of lung cancer patients will develop RILI [2,15,20]. Some of the direct and indirect radiation-induced pulmonary effects may begin within nanoseconds after radiation exposure through induction of free radicals, leading to synthetization and secretion of growth factor between a few hours and days following irradiation. The tissue undergoes progressive and dysregulated processes, i.e., inflammation-induced depletion of alveolar surface cells, infiltration of inflammatory cells into the interstitial space, exudative response, and fibrotic changes [2,21,22,23]. Radiation-induced lung disease (RILD) following radiotherapy is separated into two phases—an acute phase (lung infections and inflammations, pneumonitis) during the first 6-months and a permanent phase (pulmonary fibrosis) >6-months post-RT, but in spite of that, RILD is a dynamic process, and it is still unclear how the acute and chronic phases relate to each other [24]. The acute phase is characterized by infiltration and accumulation of inflammatory immune cells in the lung alveoli from the vascular side, loss of type I pneumocytes, and increased capillary permeability, resulting in an interstitial and alveolar edema [15,16,25]. The late phase of RILI is characterized by endothelial damage, fibrin proliferation, and disproportionate extracellular matrix development, leading to impaired gas exchange [26,27].

Radiation-induced pneumonitis (RP) is considered to be the most serious dose-limiting complication of RT. The severity of RP depends on multiple factors, including treatment factors (dosimetric parameters—total radiation dose, number of fractions, volume of irradiated parenchyma), physiologic factors (age, gender), and genetic factors (genetic variants that confer radiosensitivity) [2,15,25,28]. Typical symptoms of RILI include dry cough, dyspnea, low-grade fever, shortness of breath, chest pain and discomfort, the radiologic findings on chest X-ray and computed tomography (CT) scan are typical symptoms of RILI [29]. Clinical research showed that interleukin (Il)-1, Il-6, tumor necrosis factor (TNF)-α, transforming growth factor (TGF)-β and platelet-derived growth factor (PDGF) are associated with the occurrence of radiation pneumonitis [30].

Radiation-induced lung fibrosis (RILF), i.e., the late phase of RILI, is characterized by a tissue repair response triggered by chronic inflammation that is a culprit of excessive free radical production and chronic changes in immunological mediators. Long term upregulation of TGFβ, Il-1, or TNFα and continuous free radical production are leading to the formation of morphological changes in the structure of inter/intracellular spaces. These changes are characterized by fibroblast replication with excessive extracellular matrix deposition and are resulting in reduced lung compliance and increased work of breathing [31]. The clinical symptoms include progressive dyspnea, deterioration of pulmonary function, and interstitial fluid accumulation, leading to the respiratory failure and even a death. Pulmonary fibrosis may be divided into three specific phases. The first phase usually occurs in a few days to 3–4 weeks, sometimes up to 2–3 months after the RT, and it is characterized by pneumocytes type II damage, with the release of significant amounts of surfactant. Usually, the first phase is not detected in histopathological or radiological examination. The next phase, i.e., exudative phase, occurs between 3 to 6 months after RT. Histopathological analysis of the lung alveoli reveals the leukocyte–macrophage reactions and thickening of the intercellular space due to the fibrin fibers deposition. Characteristic physical symptoms of this phase are cough, shortness of breath, chest pains, a mild fever, and tachycardia. The third phase, acute pulmonary fibrosis, occurs a few months after radiotherapy. Lung fibrosis is characterized by the increasing inflammation, fibroblasts migration, and collagen deposition in pulmonary tissue. These changes reduce lung capacity and lead to a significant impairment of gas exchange [32,33,34].

Post-radiation side effects are correlated with chronic inflammatory diseases and are characterized by common, epigenetic pathogenesis basis. Histone acetylation regulates activation and inhibition of inflammatory genes known to play crucial roles in chronic inflammatory diseases [31]. The BET (bromodomain and extraterminal domain) family proteins are epigenetic reader proteins that recognize acetylated chromatin (the histone acetylation code of epigenetic modifications). These proteins are involved in the regulation of inflammatory cytokine genes expression in macrophages and take a part in cancer development [31,35]. Modulation of the BET proteins genes expression may drive the inhibition of proinflammatory responses, so BET proteins inhibition via JQ1 targeting is a novel therapeutic strategy in fibrosis treatment. In patients treated with combination of RT and JQ1, significant attenuation of the interstitial septal thickening, inflammatory infiltration, and fibrotic nodules in the alveolar structures and pulmonary parenchyma were observed. Moreover, decreased expression of collagen I and TGFβ indicates a radioprotective role of JQ1 [31].

Tissue injury induces inflammatory and repair responses that involve myriad interactions in epithelial-mesenchymal communication. Epithelial and inflammatory cells release profibrotic mediators such as TGFβ, PDGF, Il-1β and TNFα [7]. Ionizing radiation (IR) leads to massive free radical production and DNA damage that triggers cell death through apoptosis or necrosis. In case of cell death, pro- and anti-inflammatory cytokines and chemokines are secreted, and their overproduction generates reactive oxygen species (ROS) and nitric oxygen (NO). Chronic oxidative damage stimulates collagen production, leading to the tissue stiffness observed in RILF [16]. Cytokines such as Il-1, Il-4, Il-6, Il-8, Il-13, Il-33, TNFα, or TGFβ can be used to monitor the course and effectiveness of RT, as well as to determine the risk of side effects occurrence. There are a number of potential, specific markers, characteristic for pulmonary alveoli damage.

Other obstacles in the course of RT treatment are diverse radiosensitivity and radioresistance. However, there is substantial evidence pointing at microRNAs’ (miRNAs) involvement in the RT response. miRNAs are a class of small, single-stranded, non-coding RNAs that post-transcriptionally regulate gene expression and influence signaling pathways that could alter a series of cellular functions, including the response to IR [36,37,38]. miRNAs may be an appropriate tool to (1) profile the tumor’s radioresistance before treatment delivery; (2) monitor the response in the course of treatment and, on this basis, select intensification strategies; and (3) define the final response to the therapy along with risks of recurrence or metastization [36].

## 3. Promising Circulating Biomarkers in Radiotherapy Monitoring

### 3.1. Pro- and Anti-Inflammatory Cytokines

#### 3.1.1. Transforming Growth Factor-β

The TGFβ family is a group of pleiotropic growth factors (TGFβ isoforms, anti-Mullerian hormone, bone morphogenic proteins—BMPs) that activate the signal transduction cascade involved in carcinogenesis and tumor progression. These cytokines are a master regulator of lung development, inflammation, and injury-repair processes, and their role depends largely on their expression [10,39]. In mammalians, there are three isoforms present—TGFβ1, TGFβ2, and TGFβ3, all of which activate the same receptors and share some functions [40]. The human TGFβ1 gene is located on chromosome 19q13.1-13.39. There is an association between rs1800469 (C-509T) and rs1800470 (T869C) polymorphisms and susceptibility of radiation pneumonitis. rs1800469 polymorphism is associated with higher TGFβ production and is more prevalent in smokers [7,41].

The TGFβ is a polypeptide synthesized as a large latent precursor molecule, activated by interactions with integrins after the injury of pulmonary epithelium or endothelium. Integrin αvβ6-mediated activation is the most recognized and studied mechanism in pulmonary fibrosis. Furthermore, the TGFβ may be activated by matrix metalloproteinases (MMPs), plasmin, thrombospondin, extracellular matrix (ECM) proteins such as VCAN (versican) and ED-A-FN (extradomain-A fibronectin) or by extracellular acidification and ROS [7,42]. TGFβ’s active form binds to a complex consisting of two Type I (TGFβRl) and two Type II (TGFβR2) TGFβ receptors, which initiates an intracellular signaling cascade. TGFβ’s signaling goes through phosphorylation and activation of the intracellular receptor-regulated R-Smad (Smad2 and Smad3) proteins in fibroblasts. Activated Smads translocate to the nucleus, where activate transcription of the genes involved in fibroblast differentiation and matrix synthesis [7,31,39].

The TGFβ1 mediates cellular processes, including growth, differentiation, cell migration, chemotaxis, and apoptosis, and its expression is elevated in the airway epithelium, alveoli macrophages, airway smooth muscle cells, and fibroblasts in various pulmonary disease entities. TGFβ1 was reported as one of the most important growth factors among the molecules expressed in tissues following IR exposure. TGFβ1 is also associated with the incidence of RP and may serve as a sensitive marker of fibrinolytic changes that stimulate the differentiation of fibroblasts into myofibroblasts [10,39,43,44]. Myofibroblasts drive the cycle of increased epithelial injury (especially alveolar type II cells) and boost collagen synthesis. TGFβ1 also promotes goblet cell hyperplasia, subepithelial fibrosis, epithelial damage, and airway smooth muscle hypertrophy [39]. TGFβ1 is the most commonly used marker in the course of RILI. In the study by Wang et al. (2017), it has been shown that higher TGF-β1 2w/pre ratio (the ratio between TGFβ1 plasma level before and two weeks after radiotherapy) is associated with higher risk of RILI, and the TGFβ1 may predict RILI at 2 weeks during radiotherapy [45]. TGFβ1′s evaluation before, during, and after RT may estimate the risk of complications. The elevated levels of TGFβ1, before the implementation of the therapy, do not mean that the patient will develop RP and subsequent fibrosis; however, the persistent high level of TGFβ1 after therapy suggests the probability of radiation-induced inflammation occurrence [44,46,47].

Modulation of TGFβ1 signaling can be a future direction in treatment. Animal studies have revealed that anti-TGFβ antibodies can attenuate RILI and reduce inflammation by decreasing the level of TGFβ1. A radioprotective effect of JQ1 was also showed, which through suppressing intracellular signaling pathways, leads to weakening of RILF and collagen production reduction [31].

#### 3.1.2. Interleukins

Increased neutrophils and macrophages accumulation is observed in irradiated lung tissues and in the later RILF. Macrophages migrate from the bone marrow to the alveolar space and act as a source of numerous cytokines, including interleukin. The main proinflammatory interleukins of acute lung response are Il-1α and Il-6 [25].

Interleukins are synthesized by a large variety of cells, including monocytes, alveolar macrophages, type II pneumocytes, fibroblasts, and T lymphocytes. Interleukins may also be released from the damaged tumor cells, which have a crucial role in the immune system host defense and tumorigenic processes [15,48].

For instance, Il-6 holds effects on cellular function regulation, i.e., growth, proliferation, differentiation, metabolism, the acute-phase reaction, angiogenesis, hematopoiesis, and apoptosis. The elevated levels of Il-6 before and after RT are connected with the development of inflammation. Overproduction of Il-6 has been described in the acute radiation-induced processes, and it may be linked to the risk and occurrence of severe RP. The data suggest that Il-6 can be used as a predictive tool in the RP development. However, Il-6, similarly to other cytokines, is pleiotropic and non-specific for the radiation injury [15,43,44,46,48].

Il-8 produced by NSCLC cells plays an important role as a neutrophil-, basophil-, and T-lymphocyte-activator and chemoattractant. Data from the animal studies showed that Il-8 induces collagen synthesis and cell proliferation. Despite this, it was found that Il-8 has anti-inflammatory effect in humans [13,45,48]. Evaluation of Il-8 before RT has revealed 4 times higher levels of Il-8 in patients without inflammatory symptoms in comparison with the group of patients with symptoms [46,49]. In the study by Wang et al. (2017), lower baseline level of Il-8 was associated with higher risk of RILT. These data suggest that Il-8 may be a good predictor of the post-RT complications risk [45].

Another anti-inflammatory cytokine is Il-10, produced by macrophages and monocytes. Its main function is inflammation restraint by the inhibition of pro-inflammatory cytokines production and reduction of the antigen-presenting cells activity. Low Il-10’s concentration in the course of RT is connected with inflammation development. In the study by Arpin et al. (2005), Il-10 levels remained low in patients with RP throughout the treatment. Simultaneously, there was a consistent increase of circulating Il-10 at 2 weeks of treatment in patients without radiation pneumonitis [50].

Il-17 also has an important role in processes of lung injury induced by RT. High serum Il-17 levels at baseline of RT may indicate an increased vulnerability to RP. The study by Guo et al. (2017) was focused on the observation of Il-17’s expression levels throughout RT. Cytokine’s level peaked at 4 weeks and subsequently declined at 8 weeks after RT. There are data that suggest that treatment with IL-17 antibody alleviated RP and subsequent fibrosis, improving patients’ OS [30].

Il-4, -13, and -18 may induce inflammatory disorders as well, playing a key role in the development of RILI and fibrosis. Such circulating interleukins may be a good predictors of RT side effects and a risk of RILI. However, single determinations are not reliable, and the best option is to identify several parameters simultaneously, several times during the course of irradiation [16,48].

#### 3.1.3. Tumor Necrosis Factor-α

TNFα, pro-inflammatory cytokine produced by activated macrophages, triggers the production of other pro-inflammatory cytokines, growth factors, and acute-phase proteins [51,52]. TNFα’s immunoregulatory effects stimulate fibroblasts growth, secretion of ECM proteins, and production of collagenases and activates the cascades of other pro-inflammatory cytokines (IL-1, IL-6, and IFN). Early release of TNFα is a critical factor after lung irradiation [43,53,54]. In the study by Zhang et al. (2008), it was shown that mouse’s lungs can be protected from RILI through TNFα signaling blocking, either via knockdown or by using antisense oligonucleotides against the TNFα receptor [55]. In another study, treatment with a recombinant TNFα receptor resulted in the fibrinolytic lesions regression within damaged lungs [43]. Data points to the TNFα participation in the initial phase of RP and to a correlation between TNFα level and the occurrence of RILT [56]. However, there is no evidence on whether TNFα may be used as the predicting factor of RILT before the treatment with RT.

### 3.2. Indicators of Pneumocytes Damage

#### 3.2.1. Protein A and D of the Surfactant

Type II pneumocytes are responsible for the secretion of pulmonary surfactant. Surfactant reduces surface tension in the alveoli and facilitates alveolar expansion, permitting in this way normal gas exchange. Furthermore, surfactants regulate lung immune response and clearance of foreign particles, debris, and inflammatory material [57]. Alveolar type II pneumocytes are highly sensitive to IR injury. RILI is characterized by decreased endogenous surfactant production by type II pneumocytes and its increased degradation [58]. Surfactant insufficiency leads to the alveolar collapse and so to poor health outcomes.

Surfactant, also called surface-active agent, consists of phospholipids, carbohydrates, and proteins such as surfactant protein (SP)-A and -D. SP-D reduces surface tension at the pulmonary air-liquid interface and enhances defense as the first line of innate pulmonary immunity. Surfactant proteins stimulate macrophages to produce pro-inflammatory cytokines (TGFβ, interleukins) and ROS. Radiation-induced degradation of type II pneumocytes leads to the release of SP-A and SP-D and hence to inflammation progression. Increased permeability of pulmonary epithelial cells results in facilitated passage of SP-A and SP-D to the systemic circulation and to increased levels of circulating SPs. SP-D is a more sensitive marker of the pulmonary pathological changes than SP-A [46,59,60,61].

Numerous studies revealed elevated serum and plasma levels of SP-D in patients with RP. Sasaki et al. (2001) hypothesized that serum SP-D monitoring is a practical and useful method for the early detection of RP [62]. However, there are some limitations in patients with normal SP-D serum levels before RT [46].

#### 3.2.2. Glycoprotein Krebs von den Lugen 6

A mucin-like glycoprotein Krebs von den Lugen 6 (KL-6) is another indicator of type II pneumocytes damage and may be used in RT monitoring [63]. KL-6 antigen, also called sialylated carbohydrate antigen-6, is a high-molecular-weight glycoprotein classified as “cluster 9” according to the Third International Workshop on Lung Tumor and Differentiation Antigens. Anti-KL-6 monoclonal antibody (mAb) is considered to recognize the specific MUC1 glycopeptide sequence, which makes it a potential diagnostic and therapeutic agent. However, the precise glycan structure of the epitope recognized by anti-KL-6 mAb remains unclear [64,65].

According to immunohistochemistry and cytometry, KL-6 has been classified in the MUC1 group [66,67,68]. KL-6 is strongly expressed by type II pneumocytes and bronchiolar epithelial cells. Damaged pulmonary cells release KL-6, which makes KL-6 an indicator of interstitial lung diseases and acute lung injury. KL-6 demonstrates proliferative and anti-apoptotic effects, aiding with TGFβ1 effects, which indicates its contribution in pulmonary fibrotic processes. Therefore, at the clinical level, KL-6 is considered as useful biomarker in the determination of pulmonary fibrosis activity [59,66,69]. KL-6’s increase of at least 1.5 values on the upper limit of the reference range before RT is a marker of a high risk of complications. [46,66,68]. Serum levels of KL-6 and SP-D are good markers, with a high sensitivity for the detection of patients with a high risk of RP and RP’s severity monitoring [70]. Furthermore, serum KL-6 level significantly correlates with severity and responses to therapy in pulmonary fibrosis [63]. At this point, KL-6 is not used routinely in clinical practice, but its use is expected to increase in the future. All above described markers, their functions and usage in monitoring of radiotherapy are collected in Table 2.

## 4. MicroRNAs in Radiotherapy

microRNAs (miRNAs) are the subject of interest in various medical fields, both as a diagnostic tool and therapeutic targets. Some of the miRNAs have already passed through the phases I and II of clinical trials. Multiple studies are concerned with the radiosensitizing and radioprotective role of miRNAs in patients’ response to RT treatment [19,37]. First reports point out microRNAs’ potential as predictive biomarkers in therapy response and disease prognosis. microRNAs represent ideal markers because of their (1) specificity for the administered treatment; (2) stability in tissues, and body fluids; and (3) fast, robust, and economic expression detection [19,71].

microRNAs activity affects the response to IR by their involvement in the regulatory mechanisms of the DNA damage response (DDR), at the different levels and through (1) signaling pathways, (2) checkpoints in the cell cycle, and (3) specific repair processes that restore the single- or double-strand break (SSB, DSB) [36]. In Table 3 are presented microRNAs and their involvement in the above mentioned regulatory mechanisms. Findings of Li-Peng Jiang et al. (2017) [72] provide strong evidence that miR-21 may inhibit PD-CD4 expression and activate phosphoinositide 3-kinase (PI3K)/AKT/mTOR signaling pathways, affecting the radiosensitivity of NSCLC cells. On the contrary, the study by Yin et al. (2017) [73] has revealed that disturbance of the let-7/LIN28 double-negative feedback loop is involved in the regulation of radioresistance. Assessment of let-7 expression and its target gene—LIN28 may be used as predictive biomarkers of response to RT in NSCLC patients. Weidhaas et al. (2017) [74] reported that both of the lung cell lines (normal tissue -CLR2741-, and tumor tissue -A549-) reacted after the exposition to IR, characterizing a downregulation of all miRNAs, with one exception, let-7g, which was conversely upregulated. At the same time, down regulation of miR-9 and let-7g play a critical role in activation of NF-κB1 [75]. In addition, miR-210 appears to be a component of the radioresistance of hypoxic cancer cells. miR-210 could stabilize hypoxia-inducible factor (HIF) to promote DNA repair and activate Notch signaling pathway in angiogenesis [12,76,77]. Angiogenesis is an important factor contributing to the radioresistance of lung cancer. However, the associated mechanisms underlying radiotherapy-induced proangiogenesis are unclear.

Many studies focus on demonstrating the role of specific miRNAs in radiosensitivity and the processes standing behind it. Sun et al. (2020) [78] showed that miR-125a level varies in NSCLC cell lines with different radiosensitivities. Additionally, the authors demonstrated that miR-125a-5p upregulates apoptosis in lung cancer cells, increasing their radiosensitivity. Moreover, the expression of miR-125 is regulated by a single-nucleotide polymorphism (SNP), rs12976445, which is associated with the risk of RP [79]. Liu et al. (2013) [80] showed that inhibition of miR-21 significantly enhances the sensitivity of NSCLC cells to chemotherapeutic agents (cisplatin and docetaxel) and irradiation. In this study, authors demonstrated that overexpression of miR-21 may downregulate the expression of PTEN (tumor suppressor gene, an essential regulator of cell proliferation, differentiation, growth, and apoptosis) in NSCLC cells, making it a rational therapeutic strategy for the treatment of NSCLC in the future [80,81]. Another study revealed that miR-21 overexpression is correlated with collagen production at the radiation injury site, accompanied by the significant downregulation of several miR-21 targets, including Smad7. These findings suggest that increased miR-21 contributes to fibrotic responses observed in mesenchymal cells at the injury site through the potentiation of TGF-β signaling. Moreover, local targeting of miR-21 at the injured area may have potential therapeutic utility in mitigating RILF [82].

## 5. Future Perspective

There is little information on the predictive value of functional and biological indicators for predicting post-radiation effects, including lung inflammation in lung cancer patients treated with RT. Therefore, in the future, the introduction of markers useful in monitoring RILI is a chance to improve the clinical outcomes in patients treated with RT.

## 6. Executive Summary

RT as one of the main treatment approaches in lung cancer patients should be effective but also safe. Unfortunately, it is impossible to completely eliminate the risk side effects such as acute inflammation or lung fibrosis. Therefore, various markers, both biochemical and genetic, may be used to monitor RT. These include determination of KL-6 and SP-A and -D concentrations, which provide information about the integrity of the blood–air barrier. Elevation of their concentrations indicates type II pneumocytes damage, characteristic of the first phase of RILI and pulmonary fibrosis. Surfactant proteins are sensitive and early markers of post-radiation lung injury, which makes them good predictive parameters.

Determination of proinflammatory cytokines like TGFβ, TNFα and interleukins is also helpful in such cases. Exposure to IR leads to relatively quick appearance of proinflammatory factors in the patient’s circulating system. Cytokines multiple determinations before, during, and after RT may be useful in the prediction of complications risk level. Monitoring the course of RT is extremely important in effective treatment and for improving patients’ conditions.

Another determinant of the therapy’s effectiveness is the assessment of microRNAs expressions, which are great indicators of both radioresistance and radiosensitivity. The data indicate possibilities of microRNAs application in the RT monitoring. However, they have not been moved to the clinical setting yet.

## Figures and Tables

**Table 1 jpm-10-00072-t001:** Therapeutic strategies in non-small cell lung carcinoma (NSCLC) based upon the stages of lung cancer adapted from the 8th Edition of TNM [3,18].

Eight Edition TNM Staging System				Treatment Options
Stage IA1	T1a	N0	M0	surgery alone
Stage IA2	T1b	N0	M0	
Stage IA3	T1c	N0	M0	
Stage IB	T2a	N0	M0	<4 cm surgery alone
				>4 cm surgery followed by adjuvant chemotherapy
Stage IIA	T2b	N0	M0	Surgery followed by adjuvant chemotherapy
				There is no role of postoperative radiation therapy in patients following resection of stage I or II NSCLC with negative margins
Stage IIB	T1a-T2b	N1	M0	Patients with stage I and II disease who refuse or are not suitable candidates for surgery should be considered for radiation therapy with curative intent
	T3	N0	M0	
Stage IIIA	T1-2b	N2	M0	N0 or N1 nodes—Surgery followed by adjuvant chemotherapy
	T3	N1	M0	N2 or N3 nodes—No surgery, treatment with combined chemoradiation therapy
	T4	N0/N1	M0	The optimal treatment strategy has not been clearly defined; despite many potential treatment options, none yields a very high probability of cure; stage III is highly heterogeneous, and no single treatment approach can be recommended for all patients
Stage IIIB	T1-2b	N3	M0	
	T3/T4	N0/N1	M0	
	T3/T4	N3	M0	
Stage IVA	Any T	Any N	M1a/M1b	Use of pain medications and the appropriate use of radiotherapy and systemic therapy, which may compromise of traditional cytotoxic chemotherapy, targeted therapy, and immunotherapy depending on the specific diagnosis and molecular subtype
Stage IVB	Any T	Any N	M1c	

TNM—TNM Classification of Malignant Tumors (tumor-lymph nodes-metastasis); T1—≤3 cm surrounded by lung/visceral pleura, not involving main bronchus; T1a—primary tumor ≤ 1 cm; T1b—>1 to ≤2 cm; T1c—>2 to ≤3 cm; T2—>3 to ≤5 cm or involvement of main bronchus without carina, regardless of distance from carina or invasion visceral pleural or atelectasis or post-obstructive pneumonitis extending to hilum; T2a—>3 to ≤4 cm; T2b—>4 to ≤5 cm; T3—>5 to ≤7 cm in greatest dimension or tumor of any size that involves chest wall, pericardium, phrenic nerve or satellite nodules in the same lobe; T4—>7 cm in greatest dimension or any tumor with invasion of mediastinum, diaphragm, heart, great vessels, recurrent laryngeal nerve, carina, trachea, esophagus, spine or separate tumor in different lobe of ipsilateral lung; N0—no lymph nodes metastasis; N1—ipsilateral peribronchial and/or hilar nodes and intrapulmonary nodes; N2—ipsilateral mediastinal and/or subcarinal nodes; N3—contralateral mediastinal or hilar; ipsilateral/contralateral scalene/supraclavicular; M0—no distant metastasis; M1—distant metastasis; M1a—tumor in contralateral lung or pleural/pericardial nodule/malignant effusion; M1b—single extrathoracic metastasis, including single non-regional lymph node; M1c—multiple extrathoracic metastases in one or more organs.

**Table 2 jpm-10-00072-t002:** Featured biological markers, their functions and usage in monitoring of radiotherapy.

Biological Marker	Function in Radiation-Induced Lung Injury (RILI)	Research Studies	Conclusions	Reference
**TGFβ1**	TGFβ stimulates the differentiation of fibroblasts into myofibroblasts and promotes goblet cell hyperplasia, subepithelial fibrosis, epithelial damage, and airway smooth muscle hypertrophy	Higher TGF-β 2w/pre ratio (the ratio between TGFβ plasma level before and two weeks after RT) is associated with higher risk of RILI; the persistent high level of TGFβ after therapy suggests the occurrence of symptoms of radiation-induced inflammation	TGFβ plasma levels may identify individuals at high risk for the development of RILI	[39,43,44,45,46,47]
**Il-6**	Il-6 holds effects on the regulation of cellular functions such as growth, proliferation, differentiation, metabolism, the acute-phase reaction, angiogenesis, hematopoiesis, and apoptosis	Higher concentrations of Il-6, before and after treatment, are connected with the development of inflammation; overproduction of Il-6 in the acute radiation-induced process is associated with the risk and occurrence of severe RP	Il-6 can be used as a predictive marker of the RP development	[15,43,44,46,48]
**Il-8**	Il-8 is a neutrophil-, basophil-, and T-lymphocyte-activator and chemoattractant; Il-8 induces collagen synthesis and cell proliferation and has an anti-inflammatory effect	Lower baseline level of Il-8 is associated with higher risk of RILI (patients without inflammatory symptoms have about 4 times higher levels of Il-8 than the group of patients with the presence of symptoms)	The evaluation of Il-8 before therapy can be a good predictor for the risk of complications	[13,45,46,48,49]
**Il-10**	Il-10 downregulates inflammation by inhibiting the production of pro-inflammatory cytokines and reducing the activity of antigen-presenting cells	Levels of Il-10 are remained low in patients with RP throughout the treatment; a consistent increase of circulating Il-10 is observed at 2 weeks of treatment in patients without RP	The evaluation of Il-10 throughout the treatment may be a good predictor of RP	[50]
**TNFα**	TNFα stimulates the fibroblasts growth, secretion of ECM proteins, production of collagenases, and activation of cascades of other pro-inflammatory cytokines (IL-1, IL-6, IFN)	The early release of TNFα is a critical factor after lung irradiation; blocking of TNFα signaling via knockdown or using antisense oligonucleotides against the TNFα receptor can protect mouse lung from radiation injury; treatment with a recombinant TNFα receptor results in the regression of fibrinolytic lesions within damaged lungs	TNFα may indicate RP in its initial phase; correlation between the occurrence of RILI and the level of TNFα	[43,53,54,55,56]
**SP-A and** **SP-D**	Degradation of type II pneumocytes results in facilitated passage of SP-A and SP-D to the systemic circulation and increased levels of circulating SPs; SPs stimulate macrophages to production of pro-inflammatory cytokines (TGFβ, interleukins) and ROS	Serum and plasma levels of SP-D are elevated in patients with RP	Serum SP-D monitoring is a practical and useful method for the early detection of RP	[46,59,60,61,62]
**KL-6**	KL-6 demonstrates proliferative and anti-apoptotic effects and contributes in pulmonary fibrotic processes	An increased level of KL-6 at least 1.5 values of the upper limit of the reference range before radiotherapy correlates with a high risk of complications; serum KL-6 level correlates with severity and response to therapy in pulmonary fibrosis	Monitoring of the severity of RP; useful biomarker of pulmonary fibrosis activity	[46,59,66,68,69,70]

TGFβ—transforming growth factor β; Il—interleukin; TNFα—tumor necrosis factor α; SP—surfactant protein; KL-6—Krebs von den Lugen-6; RILI—radiation-induced lung injury; ECM—extracellular matrix; IFN—interferon; ROS—reactive oxygen species; RT—radiotherapy; RP—radiation-induced pneumonitis.

**Table 3 jpm-10-00072-t003:** Pathways and mechanisms of response to radiation damage regulated by miRNAs.

MicroRNA	Effects	Reference
PI3K/AKT and MAPK signaling pathways		
miR-21 let-7 family	overexpression of miR-21 is associated with radiation efficacy attenuation and shorter median of OS in NSCLC patients underexpression of let-7 and overexpression of LIN28 regulates proliferative capability of NSCLC cells and hence promotes resistance to RT or cisplatin treatment overexpression of let-7a decreases expression of K-Ras and is related to A549 cells radiosensitization	[72,73,74,83]
Cell-cycle progression checkpoints		
miR-21 miR-34b miR-138	underexpression of miR-21 inhibits proliferation and cell cycle progression of A549 cells; it also promotes A549 cells’ apoptosis after exposure to irradiation overexpression of miR-34b increases radio sensitivity of the p53 wild type-, and KRAS mutated-cells of NSCLC, even at low doses of RT miR-138 expression in irradiated lung cancer cell decreases the SENP1 expression, resulting in cell cycle arrest in the G1/G0 phase	[84,85,86]
Double-strand break repair		
miR-101 miR-182	miR-101 acts as radiosensitizer, its overexpression reduces the levels of DNA-PKcs and ATM, thus increasing the radiosensitivity of tumor cells knockdown of miR-182 suppresses cell proliferation and increases cell apoptosis after irradiation; unrepaired DNA damage in miR-182 knockdown cells results in cell cycle arrest	[87,88]
HIF-dependent transcriptional regulation		
miR-210	hypoxic cells are resistant to radiotherapy and chemotherapy; miR-210 is a component of the radioresistance of hypoxic cancer cells that induces and stabilizes HIF-1 through a positive regulatory loop	[76]
Inhibition of NFκB1		
miR-9	overexpression of miR-9 and let-7h inhibits NFκB1, leading to the increase of RT efficiency in lung cancer treatment	[75]

PI3K—phosphoinositide 3-kinase; MAPK—mitogen-activated protein kinase; OS—overall survival; NSCLC—non-small cell lung carcinoma; RT—radiotherapy; K-ras—Kirsten rat sarcoma; A549—culture of human lung carcinoma cell line; SENP1—sentrin-specific protease 1; G1/G0—gap 1/gap 0; DNA-PKcs—DNA- protein kinase catalytic subunit; HIF—hypoxia-inducible factor; NFκB1—nuclear factor κB1.

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
