# Peer review of "Markers Useful in Monitoring Radiation-Induced Lung Injury in Lung Cancer Patients: A Review"

_jpm, 2020, doi:10.3390/jpm10030072_

Round 1

Reviewer 1 Report

The manuscript entitled “Markers useful in monitoring radiation-in-2 duced lung injury in lung cancer patients” is an intrigue revision on the predictive value of several markers for monitoring pulmonary injury in lung cancers treated with RT.

This manuscript is well written and balance, so far it should be published after some minor revisions: 

  • Abbreviations should be checked (for example TGF β in the abstract, DNA in the introduction and so on..)
  • Misspelling should be correct
  • English should be revised
  • JPM is an international journal so far, Authors in introduction should not report Poland data. Moreover, after the first sentence, a reference should be add
  • Line 49-67 – Authors should delete or synthesize the pathogenesis of lung cancer. Rather, Authors should better describe the need to find predictive markers for cancer therapies or predictive markers of side effects (they can refer to: Catacchio I, Immune Prophets of Lung Cancer: The Prognostic and Predictive Landscape of Cellular and Molecular Immune Markers. Transl Oncol. 2018 Jun;11(3):825-835. doi: 10.1016/j.tranon.2018.04.006)
  • the Radiotherapy paragraph does not focus on the objective of the review and could be summarized and added in the introduction
  • Authors should consider that miRNA could be markers for RT (they can refer to: Cellini F, Role of microRNA in response to ionizing radiations: evidences and potential impact on clinical practice for radiotherapy. Molecules. 2014 Apr 24;19(4):5379-401. doi: 10.3390/molecules19045379)

Author Response

Dear Reviewer,

Thank you very much for all comments. In the annex is full respond to all remarks.

Kind regards,

Katarzyna Wadowska

Reviewer 2 Report

Abstract:

  1. More editing is needed in the text because some sentences read awkward. E.g., “Relation of radiotherapy with the increased risk of occurrence the side effect such as radiation induced lung toxicity and radiation induced lung fibrosis requires its monitoring to improve treatment effectiveness.”, “Complete diagnostic includes …”
  2. The abstract should be re-organized to make the rationale of the current study clearer.
  3. The second and third paragraphs are totally separated from the first paragraph in logic. The abstract should be re write.

Introduction:

  1. Why the authors spent so much words to discuss the promoting effect of smoking on lung cancer? It is far from the topic of the current study.
  2. After the discussion of smoking in relation to lung cancer, then the authors discussed about the subtypes of lung cancer. It seems the authors totally forgot about the topic of this study.

Author Response

Dear Reviewer,

I am pretty sure that you are surprised as much as I am, reading my reply to your comments. Thank you very much for your critique. That kind of remarks inspire to improve. In the annex is full respond to all comments.

Kind regards,

Katarzyna Wadowska
